# Projected Changes in Extreme Wet and Dry Conditions in Greece

**Effie Kostopoulou [1],\* and Christos Giannakopoulos [2]**

1 Department of Geography, School of Social Sciences, University of the Aegean, Mytilene, 81100 Lesvos Island, Greece
2 Institute for Environmental Research and Sustainable Development, National Observatory of Athens, 15236 Athens, Greece
\* Correspondence: ekostopoulou@aegean.gr

**Abstract:** Earth's changing climate may have different effects around the planet. Regional changes in temperature and precipitation extremes are associated with damaging natural hazards. Decreases in precipitation are expected to occur in some places at mid-latitudes, for instance the Mediterranean, which has been classified as a climate change hotspot. Droughts are among the most damaging natural hazards with severe consequences in the socio-economic sectors, the environment, and living beings. In contrast, extreme heavy precipitation events may become more frequent. This study aims to project changes in precipitation extremes and assess drought variability and change across Greece. A better knowledge of the potential changes in drought variability under climate change is vital for managing potential risks and impacts associated with dry conditions. The spatiotemporal characteristics of heavy precipitation and drought events in Greece are investigated using extreme precipitation indices such as consecutive wet/dry days, total wet-day precipitation, fraction of total wet-day rainfall, maximum daily precipitation, and heavy precipitation days. The standardized precipitation index and the standardized precipitation and evapotranspiration index are also calculated to assess seasonal dryness variability. The analysis is performed using a sub-set of high-resolution simulations from EURO-CORDEX, under two different representative concentration pathway scenarios. The results show that the region is subject to future dry conditions. Total annual precipitation is found to decrease in most of the country, with western and southern parts tending to be the most vulnerable areas. The annual precipitation is estimated to decrease by 5–20% and 5–25% (RCP4.5 and RCP8.5 respectively) toward the period 2041–2070 and by 10–25% and 15–40% (RCP4.5 and RCP8.5) toward 2071–2100. Drought-related indices reveal positive trends, particularly under the high greenhouse-gas emission scenario, with the number of consecutive dry days increasing by 20–50% and 40–80% (during 2041–2070 and 2071–2100, respectively). On the contrary, extreme precipitation events tend to decrease in the future.

**Keywords:** climate change; climate indices; Greece; extreme precipitation; drought; SPI; SPEI

## 1. Introduction

Anthropogenic climate change is expected to pose serious challenges to the availability of fresh water, especially in the water-stressed areas. Across the diverse geographical areas of Europe, there is wide variability with northern regions experiencing increases in precipitation and the southern and eastern rims experiencing high to severe water stress [1–8]. The Mediterranean basin has been recognized as one of the most vulnerable regions to climate change [9] and is likely to be exceptionally affected by future climate changes, mainly by temperature rises and water-stress increases. High temperatures lead to greater evapotranspiration and drying, which contribute to the intensity and duration of drought. The warming in the region observed since the mid-1980s was followed by a rapid acceleration in the last decade of the 20th century. Now the Mediterranean region is warming 20% faster than the global average [10–12].

According to the recent IPCC report (IPCC, 2021), the air temperature in the Mediterranean has increased by 1.5 °C above the pre-industrial level, while decreased precipitation has been observed more frequently in the north part of the basin. These changes are associated with more intense and/or longer heat waves, droughts, floods, ocean acidification, and sea-level rise [13]. The changes are expected to accelerate toward the end of the 21st century. Depending on the emission scenario, the temperature is estimated to increase within a range of 0.9–5.6 °C, whereas precipitation will likely decline by 4–22% in various areas of the Mediterranean. Heatwaves, rainfall extremes, and droughts are expected to become more frequent and/or intense [13].

Recent studies project increases in temperature extremes across the Mediterranean with particular warming in the southern part of the region [14]. Warm days and nights are found to dramatically increase, while the cold ones tend to be diminishing [15]. An increase in the intensity and duration of heatwaves toward the end of the 21st century has also been found by Molina et al. [16], according to the medium- and high-emission scenarios. The warming in the Mediterranean has been reported by Zittis et al. [17,18], while, based on the very high baseline emission scenario, they projected a 30% decrease in the wet season's precipitation in the southernmost regions of Europe. Regarding precipitation extremes, contrasted trends have been indicated between the north (increasing trends) and south (decreasing trends) regions of the Mediterranean [15,19,20].

According to UNEP [10], for 2 °C of global warming, there will be reductions by 10 to 15% in summer precipitation, while an increase of 2 to 4 °C would reduce precipitation by up to 30% in southern Europe during the warm half of the year [21–23]. In contrast, heavy precipitation events are likely to increase in various regions of the Mediterranean [24,25], even in regions with decreasing precipitation totals [26].

The fast warming and the aridity strengthening in the Mediterranean will imply several risks for ecosystems and human well-being. The wildfire season lengthens, and the fire risk gets higher, while large and severe fires are projected to occur more often because of the combined effect of the drier and warmer conditions over the coming decades [27–30]. The agricultural sector is expected to face negative impacts with decreases in crop production [10,31,32], while several other sectors, such as fisheries and aquaculture, energy, tourism, transport, and industries are also subject to important environmental risks in the Mediterranean region [10], caused by the extreme climate patterns and resulting in widespread disruptions in nature and human society [33].

Current research on climate change in the Mediterranean has shown spatial and temporal heterogeneity of the global warming impacts across the basin [34,35]. Therefore, the need for climate change information at regional and local scales is necessary to better understand the risks and sensitivity of society and the environment in different regions. This study aims to provide detailed information on potential changes in regional/local climate phenomena, in particular regarding precipitation extremes, over the eastern part of the Mediterranean for use in a climate change assessment and planning. The purpose of this work is to identify and present trends on extreme precipitation indices at local scale, under a range of possible future greenhouse-gas scenarios. The analysis focuses on 21 locations across Greece and explores changes in either heavy rainfall events or droughts during the long-term period 1971–2100 using regional climate model data. The findings of this study could provide a deep understanding of the change in precipitation patterns and water availability locally, which is of great importance to be used in domestic adaptation policies and measures for water, energy, and other impacted sectors in society. The next section provides details about the climate of the study region and describes the data and methods used. Section 3 presents the findings of the study. Time series of climate extreme precipitation indices along with their decadal trends under two emission scenarios are presented and discussed. Finally, Section 4 summarizes the results and outcomes of the study.

## 2. Materials and Methods

### 2.1. Study Region

Our study area is Greece, a Mediterranean country located in the southeastern part of Europe. This is a climate-vulnerable part of the world expected to be impacted by climate change in the context of environmental, social, and economic stresses. For instance, a warming local climate may be accompanied by more frequent, extreme, and longer heatwaves. The latter is associated with other natural disasters such as droughts and wildfires.

The Greek mainland accounts for 80% of the land area, with the remaining 20% divided among nearly 3000 islands [36]. Owing to its geographical location, diverse orography, and sea–land interactions, Greece exhibits various patterns of climate. Southern and low-land subregions enjoy a Mediterranean climate with mild and wet winters. Northern and high altitudinal regions experience a continental-type climate with cold and sometimes snowy winters. Summers are warm and dry with the exception of occasional thunderstorms. The mean annual temperature is approximately 14 °C. Figure 1 shows the interannual variability of mean temperature in Greece during the 20th century and up to present. A significant upward trend is revealed from the 1970s (0.03 °C per decade) and the rate of warming has increased, particularly since the 1990s, at about 0.05 °C per decade.

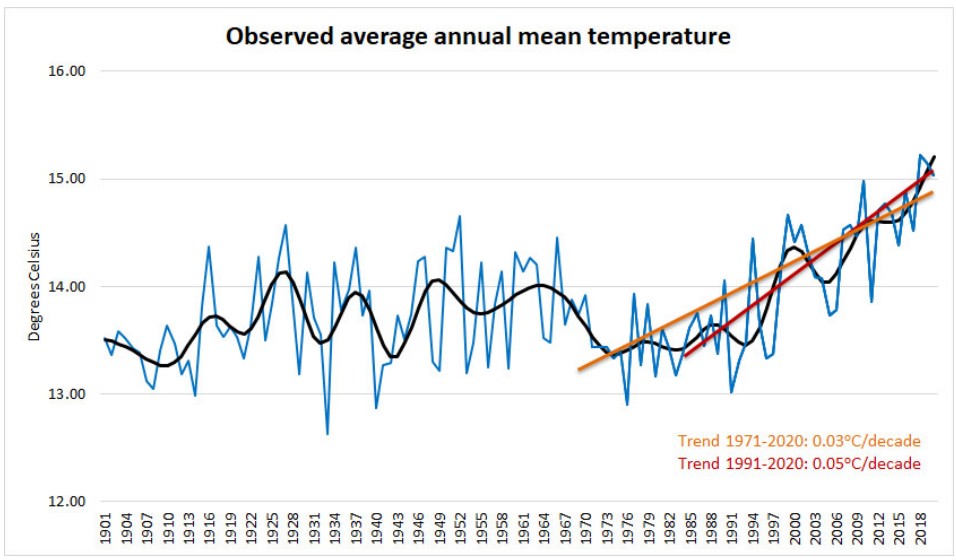

**Figure 1.** Mean annual temperature for the period 1901–2020 (blue line) and the 5-year MA (black line) and annual trends with significance over two recent periods: 1971–2020 and 1991–2020 (orange and red lines, respectively). Data source: World Bank Group, Climate Change Knowledge Portal.

Various precipitation regimes are seen throughout Greece. The mean annual precipitation is 680 mm, with annual precipitation totals over the period 1971–2000 ranging from less than 300 mm to more than 2000 mm (Figure 2). The uneven distribution of precipitation throughout the study region is due to topographic and other geographic factors at multiple spatial scales. As a result, the western elevated area of the country is the wettest (approx. 2200 mm/year in the Pindus Mountain range), while the eastern leeward areas receive considerably smaller amounts of rainfall (approx. 300 mm/year in the Cycladic Islands).

The diagram in Figure 2 shows the proportion of total annual precipitation that falls during each season. Greece receives irregular precipitation with the largest amount (approx. 40% of the annual total) occurring in winter (DJF) followed by the autumn precipitation, which accounts for the 30% of the annual total. Over the four different 30-year periods examined, we see that winter precipitation reduced over the last period, 1991–2020.

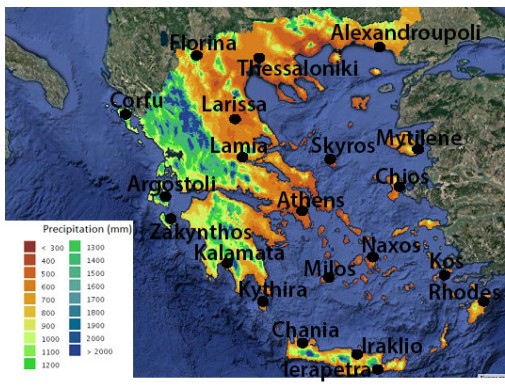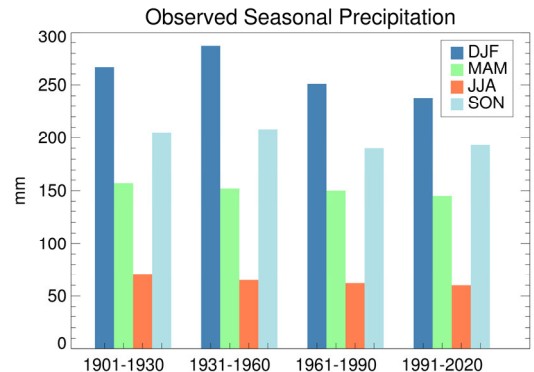

**Figure 2.** The map on the left shows the spatial distribution of mean annual precipitation throughout the study region of Greece. Blue/green (orange/red) colors represent wet (dry) subregions (source: http://climatlas.hnms.gr accessed on 25 January 2023). The locations of the stations used are marked. The diagram on the right presents the observed seasonal precipitation in different 30-year periods (data source: World Bank Group, Climate Change Knowledge Portal).

### 2.2. Climate Data

Daily time series for simulated maximum temperature (TX), minimum temperature (TN), and precipitation (PR) were used to examine extreme precipitation conditions over Greece. Station data from 21 sites throughout the study region were selected from the Hellenic National Meteorological Service. For each meteorological station location, the nearest grid point in the regional climate model dataset was identified and the time series of the examined climate variables were extracted. The regional climate model data were extracted from the EURO-CORDEX database [37]. This is the European branch of the international Coordinated Regional Climate Downscaling Experiment (CORDEX) initiative. CORDEX experiments are summarized in recent IPCC Assessment Reports, as they are often used in regional climate change impact and adaptation studies. The climate projections are utilized for the period 1971–2100, at the finer horizontal resolution of 0.11 degree (~12.5 km). The drought indices projections were estimated from a regional climate model (RCM) under two future emissions scenarios, the moderate RCP4.5 and the more extreme RCP8.5 [38]. RCP4.5 is a stabilization scenario where total radiative forcing is stabilized before 2100 by employment of a range of technologies and strategies for reducing greenhouse gas emissions. RCP8.5 is characterized by increasing greenhouse gas emissions over time, representative of scenarios in the literature leading to high greenhouse-gas concentration levels. Simulation output from the Rossby Centre regional climate model, RCA4 (driven by the MPI-ESM-LR general circulation model), was used in this study. The model data were developed by the Swedish Meteorological and Hydrological Institute [39,40] within the EURO-CORDEX initiative. RCA4 is generally found to perform well when simulating recent past climate, taking boundary conditions from the general circulation models. It has been reported that biases in large-scale circulation induce some biases in temperature and precipitation in RCA4. Subsequently, we evaluate the performance of the model over our domain of study, by comparing the simulated with observed data for the reference period 1971–2000.

### 2.3. Method and Indices

In recent decades, numerous climate indicators have been developed to examine climate-induced hydrological extremes, based on daily data of precipitation and occasionally temperature data. Most climatological studies apply the joint CCl/CLIVAR/JCOMM Expert Team on Climate Change Detection and Indices (ETCCDI) core indices [41]. The ETCCDI indices are all derived from daily precipitation data and measure aspects of frequency (e.g., days above fixed thresholds), intensity (e.g., wettest day and average daily intensity), and duration (e.g., consecutive wet and dry days) [42]. The 10 widely used ETC-

CDI precipitation-based indices are the following: consecutive dry days (CDD), consecutive wet days (CWD), total wet-day precipitation (PrcpTOT), number of heavy precipitation days (R10mm), number of very heavy precipitation days (R20mm), very wet days (R95p), extremely wet days (R99p), maximum 1 d precipitation amount (R×1day), maximum 5 d precipitation amount (R×5day), and simple daily intensity index (SDII) [43,44]. From a hydrological perspective, a drought event can be disastrous when it varies on spatial or temporal scale [45]. The standardized precipitation index (SPI), Palmer drought severity index (PDSI), and standardized precipitation evapotranspiration index (SPEI) have been defined as representative indices for drought analysis based on hydrologic and meteorological data [46].

In this study, we employ the R-software package "ClimPACT2". This software package has been developed to calculate climate indicators that are relevant for health, agriculture, water resources, and other socio-economic sectors [47]. The indicators calculated by ClimPACT2 are derived from daily temperature (TX and TN) and precipitation (PR) data. We run the ClimPACT2 for 21 locations in Greece over the period 1971–2100. The precipitation-related indices that are further discussed in this paper are presented in Table 1. The indices include the 10 core ETCCDI precipitation-based indices, along with the contributions from very wet days (R95pTOT) and extremely wet days (R99pTOT) and two indices used as measures of drought: the standardized precipitation index (SPI), which represents a precipitation deficit, and the standardized precipitation evapotranspiration index (SPEI), which is specified using precipitation and evaporation. The drought indices were calculated on time scales of 3, 6, 12, and 24 months.

**Table 1.** The definitions and units of the climate extreme indices associated with extreme precipitation used in this study.

| Index | Definition | Units |
|---|---|---|
| CDD | Maximum number of consecutive dry days (PR < 1.0 mm) | days |
| CWD | Maximum annual number of consecutive wet days (PR ≥ 1.0 mm) | days |
| R10mm | Annual number of days when PR ≥10 mm | days |
| R20mm | Annual number of days when PR ≥ 20 mm | days |
| PrcpTOT | Sum of daily PR ≥ 1.0 mm (from wet days) | mm |
| SDII | Annual total PR (PR ≥ 1.0 mm) divided by the number of wet days | mm/day |
| Rx1day | Maximum 1-day PR total | mm |
| Rx5day | Maximum 5-day PR total | mm |
| R95p | Annual total precipitation from days with PR > 95th percentile | mm |
| R99p | Annual total precipitation from days with PR > 99th percentile | mm |
| R95pTOT | Fraction of total wet-day rainfall from very wet days (100*r95p/PrcpTOT) | % |
| R99pTOT | Fraction of total wet-day rainfall from extremely wet days (100*r99p/PrcpTOT) | % |
| SPI | Standardized Precipitation Index | unitless |
| SPEI | Standardized Precipitation Evapotranspiration Index | unitless |

The standardized precipitation index (SPI) is a widely used index to identify meteorological drought on a range of temporal scales. SPI is a simple index to calculate, as monthly precipitation is the only parameter required as input, and the index can be estimated for a wide range of accumulation periods from 1 month up to 72 months [48]. The SPI indicator [49,50] quantifies observed total precipitation amounts for an accumulation period of interest, as a standardized departure (i.e., anomaly) from a selected probability distribution function that models the raw long-term historical precipitation record for that period. The SPI values for any given location and accumulation period are classified into seven different precipitation regimes (from dry to wet), as shown in Table 2. The SPI value provides a measure of the seriousness of the drought, and since it is normalized, this value is comparable from site to site and from year to year [51]. Positive SPI values indicate greater than median precipitation and negative values indicate less than median precipitation. A drought event starts when the SPI value reaches −1.0 and ends when SPI becomes positive again.

**Table 2.** SPI classification scheme.

| Anomaly | Range of SPI Values | Precipitation Regime |
|---|---|---|
| | $2.0 < SPI \leq Max$ | Extremely wet |
| Positive | $1.5 < SPI \leq 2.0$ | Very wet |
| | $1.0 < SPI \leq 1.5$ | Moderately wet |
| None | $-1.0 < SPI \leq 1.0$ | Normal precipitation |
| | $-1.5 < SPI \leq -1.0$ | Moderately dry |
| Negative | $-2.0 < SPI \leq -1.5$ | Very dry |
| | $Min \leq SPI \leq -2.0$ | Extremely dry |

The standardized precipitation evapotranspiration index (SPEI) is an extension of the SPI but includes a temperature component, allowing the index to account for the effect of temperature on drought development through a basic water balance calculation [52]. The calculation of SPEI is based on the method to calculate SPI but is modified to include the potential evapotranspiration (PET) in determining drought. SPEI employs a classification similar to SPI to define wet and dry conditions (see Table 2). The output is applicable for all climate regimes, with the results being comparable because they are standardized. With the use of temperature data, SPEI is an ideal index when looking at the impact of climate change in model output under various future scenarios. In this study, the index was calculated using the SPEI R package.

*2.4. Evaluation of the Climate Model—Metrics of Model Performance*

The present-climate simulations are tested to examine whether the models can adequately estimate the internal climate variability and reproduce the climate conditions that are associated with precipitation extremes over the study region. Several metrics were used to examine the overall agreement between the models' predictions and the observational data. Bias was first calculated as the difference between the simulated and observed long-term monthly averages. Figure 3 shows the biases regarding monthly averages of maximum/minimum temperature and monthly accumulated precipitation between the 30-year mean values of the model and observed data in 10 representative sites. The model revealed small errors and a better accuracy for temperature variables compared to precipitation. Other studies have also shown that model temperature reveals better accuracy compared to precipitation, which is highly variable in time and space [53–56]. Climate models are usually less accurate for precipitation and other complex aspects of the climate system, which are determined by various forming processes that have to be parameterized [57,58]. In our analysis, the model seems to underestimate monthly TX by up to 2.0 °C in some cases. In contrast, summer TX are overestimated in July and August in most stations (up to 2.0 °C). Similarly, monthly TN is underestimated except for the summer TN in continental regions (e.g., Kalamata and Larissa). Regarding precipitation, we see biases in both directions, with overestimations in the western (wetter) Greece (e.g., Argostoli and Corfu) and underestimations in the continental stations (e.g., Kalamata and Larissa).

Subsequently, we considered four other metrics, which were calculated from the daily time series, over the 30-year reference period (1971–2000). The mean absolute error (MAE) measures the errors between paired simulated versus observed data to estimate the accuracy of the model. The smaller the value of MAE, the better the estimation. It is common practice to also calculate the mean absolute percentage error (MAPE). MAPE is the mean absolute deviation of the compared pair divided by the actual value (here, the observation). In our study, precipitation data contain numerous zero values and thus MAPE becomes undefined. Therefore, we have also divided MAE by the mean of the observation series to calculate a weighted mean absolute error (MAE%). The lower the value for MAE, MAE%, the better the forecasting accuracy of a model. The mean squared error (MSE) was also calculated. This metric assesses the average squared distance between the observed and model values. When a model has no error, the MSE equals zero. As model

error increases, its value increases. However, squaring increases the impact of large errors. Hence, we take the square root of the MSE to obtain the root mean square error (RMSE), which is expressed in the same units and scale as the original data. RMSE is the standard deviation of the residuals (i.e., how far from the regression line data points are). RMSE equal to zero indicates a lack of error [59,60]. RMSE is a measure of how concentrated the data are around the line of best fit. Root mean square error is commonly used in climatology to verify experimental results [61]. It is useful to use MAE and RMSE together to identify errors in simulations. The RMSE will always be larger or equal to the MAE; the greater the difference between them, the greater the variance in the individual errors in the sample. If the RMSE is equal to the MAE, then the errors are of the same magnitude. The lower values indicate higher accuracy of the model. Figure 4 presents the four metrics (MSE, RMSE, MAE, and MAE%) we have calculated in this study for 10 representative sites. In summary, the skill metrics suggested that temperature variables show high agreement to the reference dataset, especially in lowland and island regions. The evaluation results obtained for model precipitation showed that they are in good agreement with the reference dataset. The model seems to better simulate precipitation in the central lowland areas of the study region.

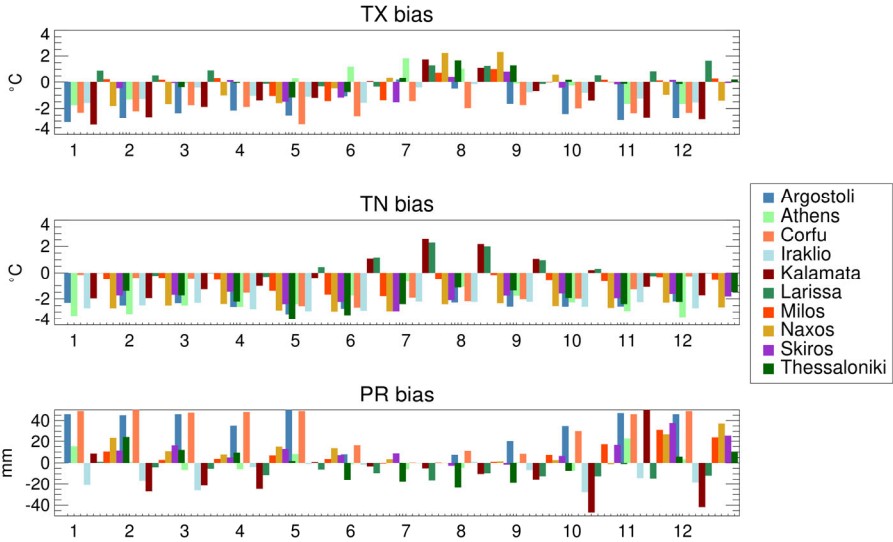

**Figure 3.** Biases in monthly statistics of daily TX, TN, and PR based on the model data for the period 1971–2000 in 10 representative Greek stations.

The Pearson correlation coefficient (r) is also often used to indicate agreement between different datasets, as a higher 'r' indicates that the two datasets have similar spatial or temporal patterns. In our analysis, the Pearson correlation coefficient values were calculated to express the degree of association between the investigated datasets, based on the daily time series of the stations. The correlation coefficients for TX and TN range between 0.75 and 0.85 at the 5% confidence level, whereas regarding precipitation, no significant correlation was identified. The results showed high correlation scores in all variables between the long-term average monthly datasets.

In addition, histograms were created to display the probability of climatic values in the model data and compare their frequency distribution against that of the observational dataset. Histograms can serve as a visual tool to check normality in a continuous variable, see data trends, and identify outliers. Moreover, histograms provide a useful statistical quality control tool, as they present the center, spread, and shape of a dataset. To evaluate the model data used, we created pairs of daily time series of TX, TN, and PR, consisting of observational and modeled data and covering a 30-year period of present climate (1971–2000) for 12 representative Greek stations. Each histogram shows the distributions of the paired series in every station. The frequency is seen on the vertical axis displaying the

number of days that a value occurs within a certain class interval (range of values). Figure 5 presents two sets of panels for TX and PR. In each set and for every station, two histograms are presented (dark and light color show the distribution of values for observations and simulations, respectively). Thus, each distribution is clearly displayed.

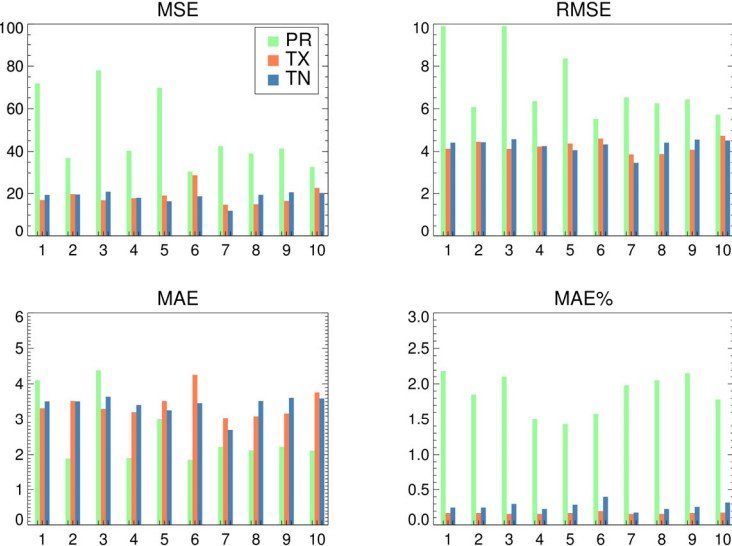

**Figure 4.** Skill metrics calculated between paired model—observed data to estimate the accuracy of the model in precipitation (PR), maximum (TX) and minimum (TN) temperature.

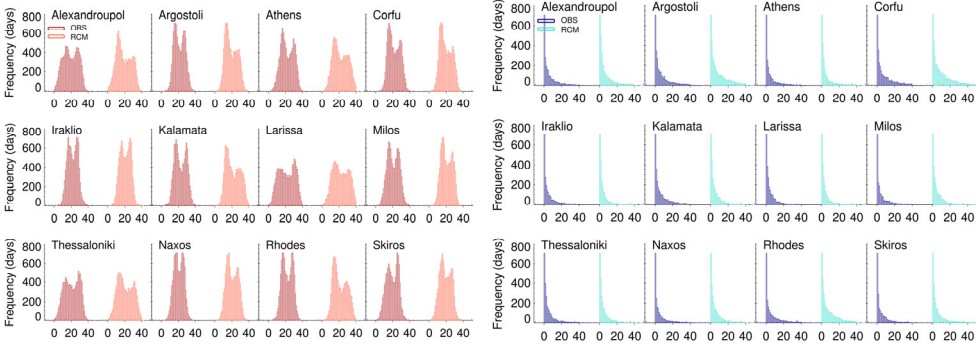

**Figure 5.** Histograms of maximum temperature observational (OBS, dark red) and model (RCM, light red) data, and precipitation (OBS, dark blue; RCM, light blue) for several Greek cities. Frequency corresponds to the number of days with temperature/precipitation values within certain classes (in °C and mm, respectively) over a 30-year period (1971–2000).

　　The histogram of the observational TX is drawn in dark-red color and of the modeled TX is drawn in light-red color. It is generally found that the RCM data adequately resemble the distribution of the observational data. In the majority of the days, maximum temperatures range between 10 and 30 degrees Celsius, while their extreme values may reach a low value of 0 °C and a high of approximately 40 °C. In all cases, we observe bimodal (doubly-peaked) distributions caused by the seasonal variations of TX. Hence, each distribution is characterized by two local maxima. The first distinct peak is approx. 15 °C (avg. maxima in cold half of the year), while the second peak is approx. 35 °C (avg. maxima of the warm half of the year). The model overestimated the number of days with the first maxima in continental areas of northern Greece (e.g., Alexandroupoli), whereas in another case, the model underestimated the number of days reaching the second maxima (e.g., Kalamata, southern Greece). Overall, the RCM captured very well the probability frequency distribution of the real data, except for some underestimations in the number of

days reaching the second maxima in island stations (e.g., Skiros and Naxos). Similar to TX, the histograms for TN (not shown) revealed that the model provides a good estimation of distribution of minimum temperature in all cases. Regarding precipitation, the observational PR is drawn in dark-blue color and of the modeled PR is drawn in light-blue color. Here, we see that the model sufficiently reproduces the distribution of precipitation in all sites, although in some cases the model counts more days in lower classes.

## 3. Results

The long-term variation or change of the climate-extreme indices are presented and discussed for the entire period of study, 1971–2100. The calculation of the indices was made for 21 Greek stations, and therefore, several graphical representations for different representative sites are displayed in every case. The linear trend line is also shown in the graphs to indicate particular tendencies for extremes to change (increase or decrease) or not. Future changes of the indices are presented based on two RCP scenarios (RCP4.5 and RCP 8.5).

### 3.1. Time Series of Climate Extreme Precipitation Indices

3.1.1. Annual Total of Precipitation in Wet Days (PrcpTOT) and Simple Daily Intensity Index (SDII)

The PrcpTOT accounts the annual amount of precipitation from wet days (precipitation of more than 1.0 mm), while SDII is defined as the ratio of annual precipitation to the number of wet days (mm/days). These indices are not directly related to extreme climate events, but they demonstrate changes in the precipitation distribution. Our analysis did not detect particular changes in SDII, meaning that daily precipitation intensity has neither changed nor is expected to change toward the end of the 21st century. Regarding PrcpTOT significant decreases are found in stations of the western and southern part of Greece under RCP4.5, and in all the territory under RCP8.5. Overall, a significant decrease in annual total precipitation is found for every station across Greece from −5.2 mm/decade up to −78 mm/decade (Table 3). Figure 6 presents the long-term (1971–2100) temporal variability of PrcpTOT in four representative stations, exhibiting strong significant negative trends under the high-emission scenario. The stations of Athens and Kalamata represent areas with drier conditions in the central and southern parts of the country, where the reduction approximates −14 mm per decade. Larger reductions were identified in western and southeastern parts of Greece as represented by Argostoli and Rhodes with reductions being around −45 mm per decade. The largest decrease is found in the station on Zakynthos (−78.1 mm/decade, Table 3). The results show that anthropogenic climate change may well influence the variability of annual precipitation in the study area during the period 1971–2100.

3.1.2. Maximum Annual Number of Consecutive Wet/Dry Days

The indices of consecutive wet days (CWD) and consecutive dry days (CDD) are related to the precipitation occurrence or absence, which is of great interest for many human activities. The interannual variation of the maximum number of CWD (precipitation 1.0 mm or more) for the entire period 1971–2100 under both scenarios is presented in Figure 7 for four representative Greek stations. Particularly, regarding the historical period (1971–2000), the annual CWD range from 7–25 days in Zakynthos, in the wetter part of western Greece, to 3–11 days in Athens, one of the drier parts of the country. The future projections (2071–2100) estimate the CWD range in Zakynthos to 5–17 days and 4–14 days under RCP4.5 and RCP8.5, respectively. Regarding Athens, the range reaches 3–7 days under RCP4.5 and 2–6 days under RCP8.5. In general, the wet-day sequences showed downward trends in all stations. However, the rates of reduction are rather low, corresponding to less than one day per decade. The area is likely to experience large decrease in wet days by the end of this century under the RCP8.5, especially in western and southern stations. For instance, the number of CWD is seen to decrease significantly ($p < 0.05$), by approx.

−0.4 days per decade (Table 3), in the stations of Chania, Rhodes, Kithira, Argostoli, and Zakynthos.

**Table 3.** Decadal trends in annual extreme precipitation indices under RCP4.5 and RCP8.5 for 21 Greek stations during the period 1971–2100 (numbers in bold indicate statistically significant slopes, $p < 0.05$).

| Stations | PrcpTOT | | SDII | | CWD | | CDD | | R10mm | | R20mm | |
|---|---|---|---|---|---|---|---|---|---|---|---|---|
| | R4.5 | R8.5 | R4.5 | R8.5 | R4.5 | R8.5 | R4.5 | R8.5 | R4.5 | R8.5 | R4.5 | R8.5 |
| Alexandroupoli | −3.28 | **−14.2** | **0.08** | **0.1** | −0.06 | **−0.17** | **1.24** | **2.92** | −0.07 | **−0.38** | 0.03 | −0.09 |
| Argostoli | **−17.4** | **−45.8** | −0.01 | **−0.09** | −0.07 | **−0.35** | **1.29** | **2.51** | **−0.86** | **−1.79** | **−0.34** | **−0.95** |
| Athens | −3.81 | **−10.6** | 0 | 0.01 | **−0.12** | **−0.12** | 0.42 | **3.67** | −0.16 | **−0.3** | −0.05 | −0.08 |
| Chania | **−12.9** | **−29.5** | −0.02 | −0.04 | **−0.21** | **−0.41** | **1.36** | **3.88** | **−0.4** | **−1** | −0.15 | **−0.31** |
| Chios | −8.26 | **−22.9** | 0.02 | 0.04 | −0.1 | **−0.27** | **1.16** | **3.69** | **−0.37** | **−0.9** | −0.11 | **−0.35** |
| Corfu | **−17.2** | **−36.4** | 0.01 | 0.02 | −0.04 | **−0.2** | 0.96 | 1.38 | **−0.63** | **−1.46** | −0.26 | **−0.53** |
| Florina | −2.05 | **−8.27** | 0.02 | **0.05** | −0.04 | −0.08 | **1.18** | **2.3** | −0.06 | **−0.22** | 0.02 | 0 |
| Ierapetra | **−7.1** | **−17.1** | −0.01 | −0.04 | **−0.14** | **−0.18** | 0.72 | **2.4** | **−0.22** | **−0.52** | −0.09 | **−0.18** |
| Iraklio | −4.67 | **−9.59** | 0.02 | 0.01 | −0.1 | **−0.14** | **1.49** | **5.74** | −0.1 | **−0.22** | 0 | −0.06 |
| Kalamata | −3.07 | **−14.1** | 0.03 | −0.02 | 0.04 | **−0.2** | 0.33 | **2.35** | −0.01 | **−0.4** | 0.01 | −0.11 |
| Kithira | **−25.1** | **−43.6** | **−0.12** | **−0.13** | **−0.2** | **−0.38** | **1.81** | **4.11** | **−0.92** | **−1.64** | **−0.51** | **−0.73** |
| Kos | **−9.96** | **−21.5** | −0.01 | 0 | **−0.18** | **−0.25** | 0.43 | **3.52** | **−0.39** | **−0.83** | −0.08 | **−0.25** |
| Lamia | −1.45 | **−6.03** | 0.03 | **0.06** | −0.07 | **−0.1** | **1.34** | **4.43** | 0.07 | −0.01 | 0.01 | −0.02 |
| Larissa | 1.19 | **−6.7** | **0.1** | 0.05 | −0.02 | −0.06 | **1.38** | **3.04** | 0.05 | **−0.17** | 0.08 | 0.02 |
| Milos | **−8.04** | **−15.2** | 0.01 | 0.04 | **−0.17** | **−0.23** | **1.67** | **4.27** | −0.16 | **−0.49** | −0.09 | **−0.12** |
| Mytilene | **−15.9** | **−36.0** | 0.06 | 0.06 | −0.09 | **−0.31** | **1.37** | **3.41** | **−0.63** | **−1.5** | **−0.37** | **−0.66** |
| Naxos | −5.13 | **−15.0** | 0.02 | 0.01 | **−0.1** | **−0.18** | 0.56 | **3.26** | −0.19 | **−0.5** | −0.07 | **−0.13** |
| Rhodes | **−27.6** | **−46.5** | −0.08 | −0.08 | **−0.25** | **−0.39** | **1.69** | **4.55** | **−1.01** | **−1.74** | **−0.49** | **−0.73** |
| Skiros | −4.62 | **−10.7** | 0.05 | 0.03 | −0.06 | −0.07 | **1.34** | **3.63** | −0.21 | **−0.42** | −0.02 | −0.07 |
| Thessaloniki | −0.75 | **−5.19** | **0.08** | **0.1** | −0.05 | −0.08 | **1.23** | **2.1** | 0.03 | −0.06 | 0.08 | 0.07 |
| Zakynthos | **−34.7** | **−78.1** | −0.05 | **−0.18** | −0.14 | **−0.35** | **1.06** | **2.24** | **−1.08** | **−2.37** | **−0.77** | **−1.63** |

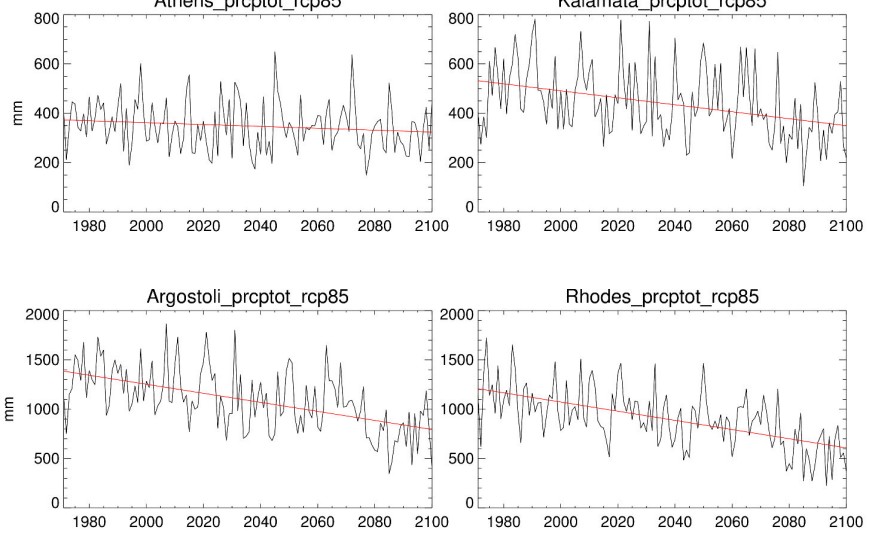

**Figure 6.** Annual sum of precipitation in wet days, where precipitation is at least 1 mm (PrcpTOT) during 1971–2100 in four Greek stations based on RCP8.5.

In contrast, the consecutive dry days are expected to increase in the future over Greece. All stations exhibited increasing trends in CDD, implying an overall tendency toward a drier future in the area. As proof, we refer to the results of two stations, namely Zakynthos and Athens, which are characterized by wet and dry climate conditions, respectively. Regarding the historical period, the annual number of CDD in Zakynthos ranges between 21–74 days and in Athens between 40–158 days. The future estimations reveal increased ranges of 26–116 days and 24–156 days for Zakynthos and 46–173 days and 49–239 days for Athens, under RCP4.5 and RCP8.5, respectively. Overall, the number of CDD is seen to

significantly increase by approx. +1.5 days per decade (Table 3) in many stations under RCP4.5. Moreover, under the RCP8.5 scenario, the positive trends are all statistically significant with the stations of Iraklio, Rhodes, Lamia, Milos, and Kithira revealing lengthening in dry spells by up to 4–5 days per decade. Figure 8 shows the interannual variability in CDD from 1971 to 2100 and the rising signal of the index in eight stations located in different parts of the country.

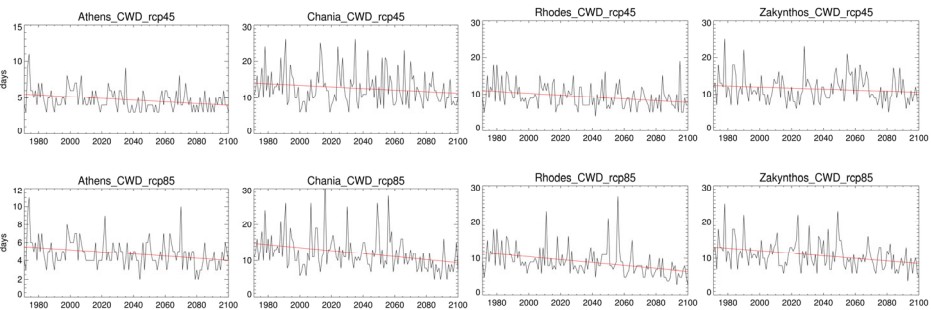

**Figure 7.** Consecutive wet days (CWD) during 1971–2100 in four Greek stations based on RCP4.5 (**top row**) and RCP8.5 (**bottom row**).

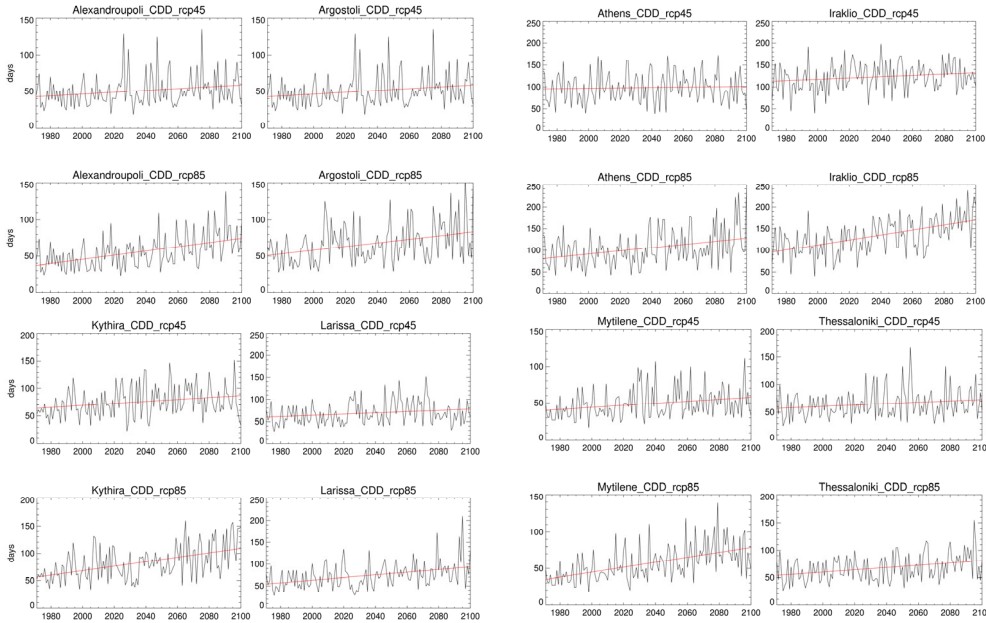

**Figure 8.** Consecutive dry days (CDD) during 1971–2100 in eight Greek stations based on RCP4.5 (**first** and **third** rows) and RCP8.5 (**second** and **fourth** rows).

### 3.1.3. Annual Number of Days with Precipitation Not Less Than 10 mm (R10mm) or 20 mm (R20mm)

The indicators of heavy (R10mm) and very heavy (R20mm) precipitation days reveal similar results. Both indices tend to decrease with statistically significant trends at the 5% level, mainly found in R10mm for RCP8.5 (Figure 9, R20mm not shown). The reductions reach up to −2.4 days and −1.6 days per decade under RCP8.5. Particularly, the number of days with heavy and very heavy precipitation exhibit larger decreases in the western and southern parts of the country (e.g., Argostoli, Zakynthos, Ierapetra, and Rhodes). Figure 9 illustrates the temporal variability of the annual number of days with precipitation not less than 10 mm (R10mm) during 1971–2100 (under RCP8.5) in four representative stations located in different parts of the country. Overall, the frequency of heavy precipitation days tends to significantly decrease in Zakynthos up to −2.4 days/decade, as well as in Argostoli (−1.79 days/decade) and Rhodes (−1.74 days/decade).

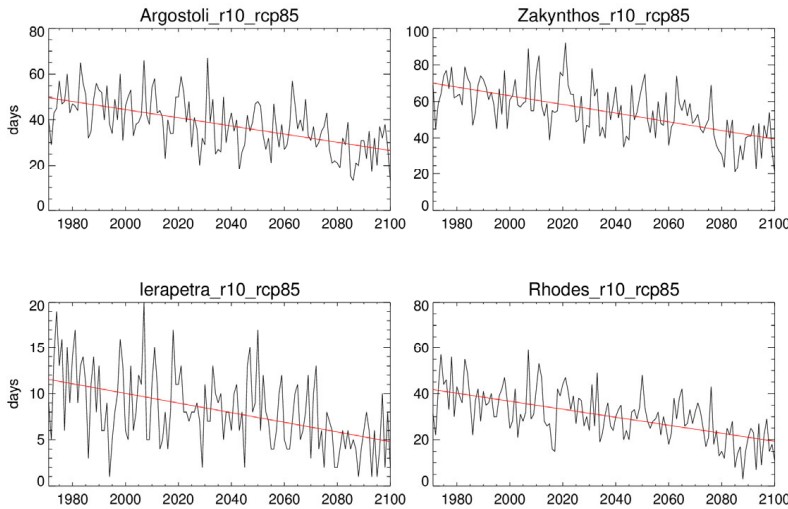

**Figure 9.** Annual number of days with precipitation not less than 10 mm (R10mm) during 1971–2100 in four Greek stations, based on RCP8.5.

### 3.1.4. Annual Maximum 1-Day (Rx1day) and 5-Day (Rx5day) Precipitation

The maximum 1-day and 5-day precipitation indices showed weak nonsignificant trends across the study region. Some indicative cases of the temporal variability of the two indices, under RCP8.5, are presented in Figure 10. The stations' temporal patterns resemble one another and present large interannual variability. Toward the end of the 21st century, the maximum precipitation in 1 day tends to reduce in many sites (e.g., Thessaloniki, Iraklio, and Argostoli), although some distinct exceptions are projected. Downward trends are observed for RX5day over 1971–2100 in most examined sites. The highest negative changes are found over the western and southern parts of Greece (e.g., Zakynthos and Rhodes).

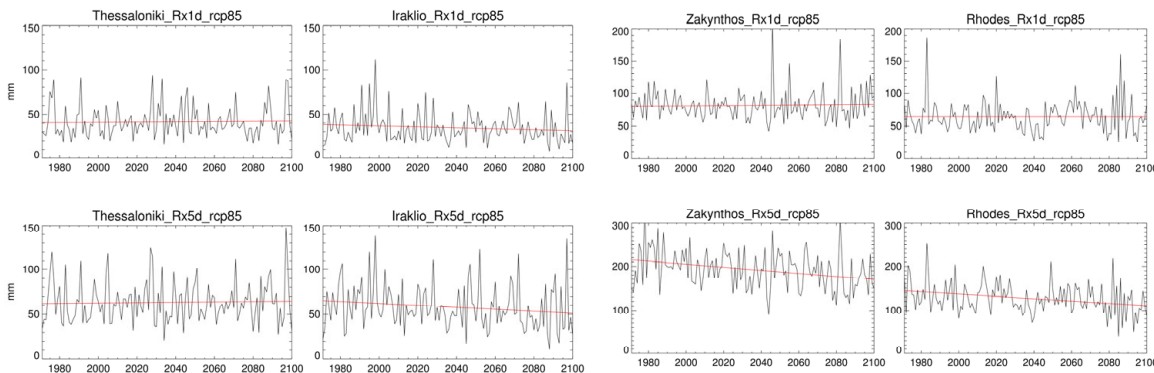

**Figure 10.** Annual maximum 1-day (Rx1d) and 5-day (Rx5d) precipitation totals during 1971–2100 in four Greek stations based on RCP8.5.

### 3.1.5. Annual Total Precipitation from Days with PR > 95th (R95p) and PR > 99th (R99p) Percentile

The indices R95p and R99p were also calculated to examine changes in intense precipitation events. The extreme precipitation is estimated by the annual accumulated amount of precipitation from days with precipitation above the 95th and 99th percentiles, with respect to the reference period (1971–2000). In much of the country, the trends are not significant mainly because of the large interannual variability of precipitation. Consistently with previous findings, significant decreases tend to occur in Kythira, Rhodes, and Zakynthos (Figure 11).

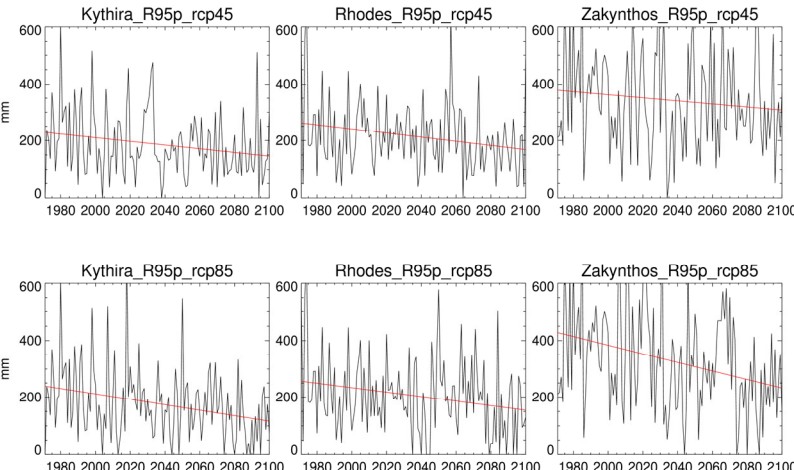

**Figure 11.** Annual total precipitation from days with precipitation exceeding the 95th percentile (R95p) under both emission scenarios.

3.1.6. Annual Fraction of Total Wet-Day Rainfall from Very Wet Days (R95pTOT) and Extremely Wet Days (R99pTOT)

Results similar to the previous results were also obtained from indicators reflecting the contribution of very wet (R95pTOT) and extremely wet days (R99pTOT). The indices produced large interannual variability in these extreme events and led to small and non-significant trends over the long-term period. Some significant results were related to upward trends of about 0.65 to 0.99% per decade (Figure 12).

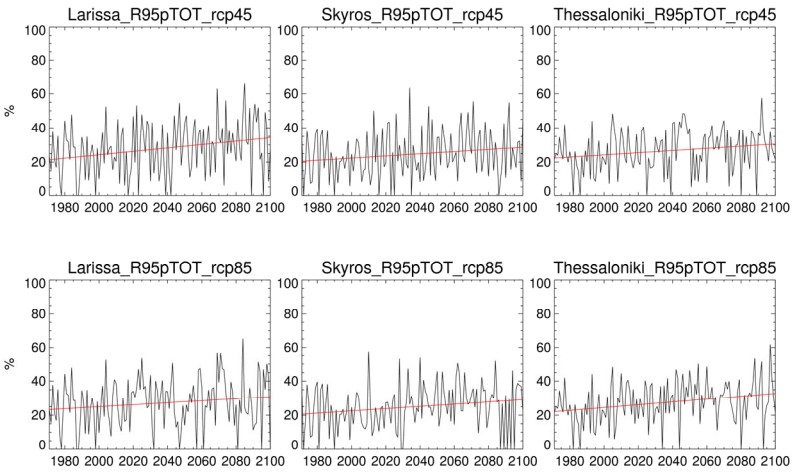

**Figure 12.** Fraction of the total wet-day rainfall from very wet days (R95pTOT) during 1971–2100 in four Greek stations based on RCP8.5.

### 3.2. Trends in Climate Extreme Precipitation Indices

Linear trends were estimated based on Sen's slope to detect long-term changes of the extreme precipitation indices over time. The decadal rates of change under both emission scenarios for the 21 examined locations in Greece are summarized in Tables 3 and 4. These patterns of change provide understanding of how local climate features may change in future decades. We further identify those parts of the study region that reveal significant changes in extreme precipitation indices. The trend values in bold are significant at a 95% confidence level. Overall, the results show a tendency toward drier conditions. Decreases in PrcpTOT are seen in all stations under RCP4.5. The largest magnitudes among trends are found to be statistically significant. RCP8.5 projects greater and statistically significant negative trends in all sites. SDII does not show particular rates of change over time. The CDD shows increasing trends in all cases. Of particular interest are the significant positive

CDD trends under RCP8.5, which vary from +2 to +6 days per decade. The greater changes mainly refer to the southern part of the country. The CWD, R10mm, and R20mm show negative trends across the study area. Especially under the RCP8.5, wet days tend to reduce in all stations, whereas decreasing trends are revealed also for extreme rainfalls (indicated by R10 and R20), particularly in western and southern parts. Regarding Rx1day, Rx5day, R95p, and R99p, few significant changes are detected. For instance, negative trends in Rx5day and R95p are found in western and southern stations (e.g., Zakynthos, Kithira, Ierapetra, and Rhodes). R95pTOT indicates small positive trends. Significant trends in R95pTOT are seen in continental parts of Greece (e.g., Florina, Larissa, and Thessaloniki). The most significant trends in extreme precipitation indices occur in PrcpTOT, CDW, CDD, and R10mm. Figure 13 presents the spatial distribution of trends in these four indices along with R95p. The dark brown symbols (circles) indicate trends with higher magnitudes, while the yellow symbols indicate lower magnitude trends. Filled circles represent statistically significant trends. In summary, positive significant trends occur in CDD, whereas the remaining symbol maps present negative significant trends in PrcpTOT, CDW, and R10mm, excluding R95p, where trends are negative but significant only in few stations in western or southern Greece.

**Table 4.** Same as Table 3 for the remaining precipitation indices.

| Stations | Rx1day | | Rx5day | | R95p | | R99p | | R95pTOT | | R99pTOT | |
|---|---|---|---|---|---|---|---|---|---|---|---|---|
| | R4.5 | R8.5 | R4.5 | R8.5 | R4.5 | R8.5 | R4.5 | R8.5 | R4.5 | R8.5 | R4.5 | R8.5 |
| Alexandroupoli | 0.09 | −0.39 | 0.54 | −0.78 | −0.35 | −1.51 | 0.07 | −0.39 | 0.11 | 0.28 | 0.03 | 0.09 |
| Argostoli | 0.63 | 0.23 | −0.29 | −1.35 | 1.09 | **−7.24** | 0.27 | 1.09 | **0.42** | 0.09 | 0.12 | **0.37** |
| Athens | 0.45 | −0.44 | 0.07 | −0.82 | −0.39 | −1.93 | 1.14 | −0.81 | 0.06 | 0.05 | 0.28 | −0.17 |
| Chania | −0.07 | −0.1 | −0.56 | −1.45 | −3.81 | −5.79 | −0.69 | −0.5 | −0.1 | 0.07 | 0.09 | 0.24 |
| Chios | −0.28 | 0.18 | −0.95 | −0.51 | 1.15 | −0.09 | 0.09 | 0.72 | 0.36 | 0.51 | 0.07 | 0.23 |
| Corfu | −0.12 | 0.03 | −0.7 | −0.87 | −1.14 | −1.65 | −1.77 | 1.34 | 0.17 | 0.38 | −0.03 | 0.25 |
| Florina | **0.95** | 0.12 | 0.85 | −0.24 | 1.61 | −0.03 | **2.07** | 0.87 | **0.57** | 0.31 | **0.62** | 0.3 |
| Ierapetra | 0.12 | −0.64 | −0.06 | **−1.71** | −1.82 | **−4.43** | −0.26 | −0.78 | 0.03 | −0.12 | 0.06 | 0.04 |
| Iraklio | −0.26 | −0.56 | **−1.16** | −1.11 | −0.79 | −2.28 | −0.52 | −0.68 | 0.26 | −0.11 | 0.04 | −0.04 |
| Kalamata | 0.34 | −0.17 | −0.14 | −0.62 | 0.44 | −2.41 | −0.23 | −1.13 | 0.37 | 0.31 | 0.1 | −0.09 |
| Kithira | −0.19 | −0.98 | −1.56 | **−2.21** | **−6.42** | **−9.49** | −1.28 | −2.12 | −0.14 | −0.07 | 0.02 | 0.01 |
| Kos | −0.11 | 0.17 | −0.67 | −1.22 | −2.38 | −2.64 | 0.53 | 0.85 | −0.12 | 0.14 | 0.14 | 0.36 |
| Lamia | 0.46 | 0.05 | 0.43 | 0.61 | 0.13 | −0.51 | −0.11 | −0.27 | 0.13 | 0.23 | −0.01 | 0.03 |
| Larissa | 0.78 | 0.05 | 0.81 | −0.23 | **3.65** | 0.28 | **2.6** | 0.6 | **0.99** | 0.59 | **0.69** | 0.36 |
| Milos | −0.15 | 0.62 | −0.19 | −0.09 | −1.12 | −0.79 | −1.35 | 1.41 | 0.18 | 0.34 | −0.11 | 0.47 |
| Mytilene | 0.35 | 0.94 | −0.87 | −0.28 | 1.19 | −1.17 | 1.56 | 2.24 | 0.34 | **0.54** | 0.18 | **0.45** |
| Naxos | 0.04 | −0.13 | −0.43 | −0.37 | 0.29 | −0.85 | 1.16 | 1.19 | 0.3 | 0.34 | 0.3 | 0.38 |
| Rhodes | −0.37 | −0.01 | −1.69 | **−2.61** | **−7.2** | **−7.9** | −1.81 | −1.62 | −0.11 | 0.15 | −0.03 | 0.22 |
| Skiros | **1.26** | 0.57 | 1.27 | 0.49 | 1.69 | 1.18 | 2.36 | 1.96 | **0.63** | **0.66** | **0.59** | 0.52 |
| Thessaloniki | −0.36 | 0.13 | −0.45 | 0.21 | 2.44 | 2.13 | −0.06 | 0.14 | **0.63** | **0.83** | 0.08 | 0.1 |
| Zakynthos | 0.5 | 0.24 | **−2.41** | **−3.52** | −5.45 | **−15.1** | −0.33 | −2.41 | 0.09 | 0.01 | 0.12 | 0.16 |

*3.3. Time Series of Drought Indices*

3.3.1. Temporal Analysis of SPI-3, SPI-6, SPI-12, and SPI-24

The standardized precipitation index (SPI) is a widely used statistical estimator of precipitation deficit and meteorological drought for any time scale. SPI calculates the precipitation amount in a station during monthly periods of time and compares it with the expected precipitation based on the long-term precipitation received at this station over the same period. Usually, SPI is calculated for monthly time periods of 1 to 48 months. Short periods (i.e., based on SPI-1 to SPI-3) indicate changes in soil moisture. Medium-term trends in precipitation patterns (i.e., SPI-6 to SPI-12) reflect impacts in stream flow and reservoir levels, while long periods (i.e., SPI-24) are examined for potential shortages in groundwater recharge. In this study, we discuss the variation and trends of SPI-3, SPI-6, SPI012, and SPI-24 over the period 1971–2100, in comparison with the specific n-month period from the 30-year historical record of 1971–2000.

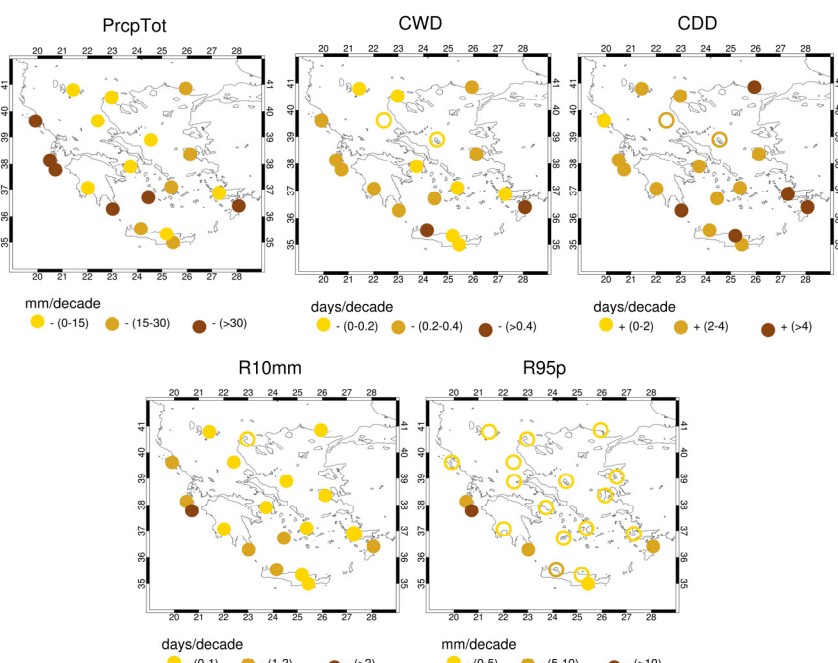

**Figure 13.** Spatial distribution of trends in extreme climate indices during 1971–2100 over Greece based on RCP8.5. The brown (yellow) circles indicate higher (lower) trends. Filled circles represent statistically significant trends.

The temporal fluctuation of SPI for five representative Greek stations, over the period 1971–2100 under RCP 8.5, is presented in Figure 14. The changes in SPI are drawn in two-colored bars, with blue representing high and red low SPI values (i.e., wet and dry years, respectively). All stations in every SPI time scale reveal extreme dry conditions (SPI < −2), mainly during the last 30-year study period. Regarding SPI-3, extreme dry conditions are anticipated to occur at the middle, as well as at the end, of the 21st century. Furthermore, we examined seasonal three-month SPI (not shown). More specifically, the winter SPI shows that rainfall amounts tend to reduce toward the end of the 21st century, particularly in western and southern parts of the country (e.g., Argostoli and Chania). Low SPI values are also found for Alexandroupoli and Athens. It is worth noting that the station of Larissa, in the central part of the country, did not show a downward trend in the SPI-3 of February, but it did for the SPI-3 of January, denoting lower rainfall from November to January. As regards the three-month SPI of autumn, we did not detect clear rates of change except for the tendency for negative SPI values in western and central parts of Greece (e.g., Argostoli and Larissa). Significant change from the middle to the end of the 21st century in three-month SPI is detected during spring, when most stations reveal negative values corresponding to drier-than-normal periods. Finally, during summer, when rainfall rarely occurs over Greece, the second half of the 21st century shows moderate to extreme dry years, except in particular cases that are mainly due to seasonal thunderstorm events. The temporal patterns of SPI-6 were found similar to SPI-3. Extreme dry years are expected in all stations, mainly after the 2070s. The deficiency in precipitation may lead to meteorological droughts. Therefore, the standardized precipitation index for 12- and 24-month accumulation periods displays the longer-term drought, summarizes the shortages of shorter-period rainfall, and provides information on the risks of drought impact for agriculture and water resources. The longer SPIs are indicative of trends in precipitation patterns that may take place. Here, we found that hydrological droughts are expected in most stations. The annual droughts intensify in the middle and become more severe by the end of the 21st century. The temporal SPI-24 patterns are similar to those of SPI-12, with similarly severe values and higher persistence of extreme dry conditions.

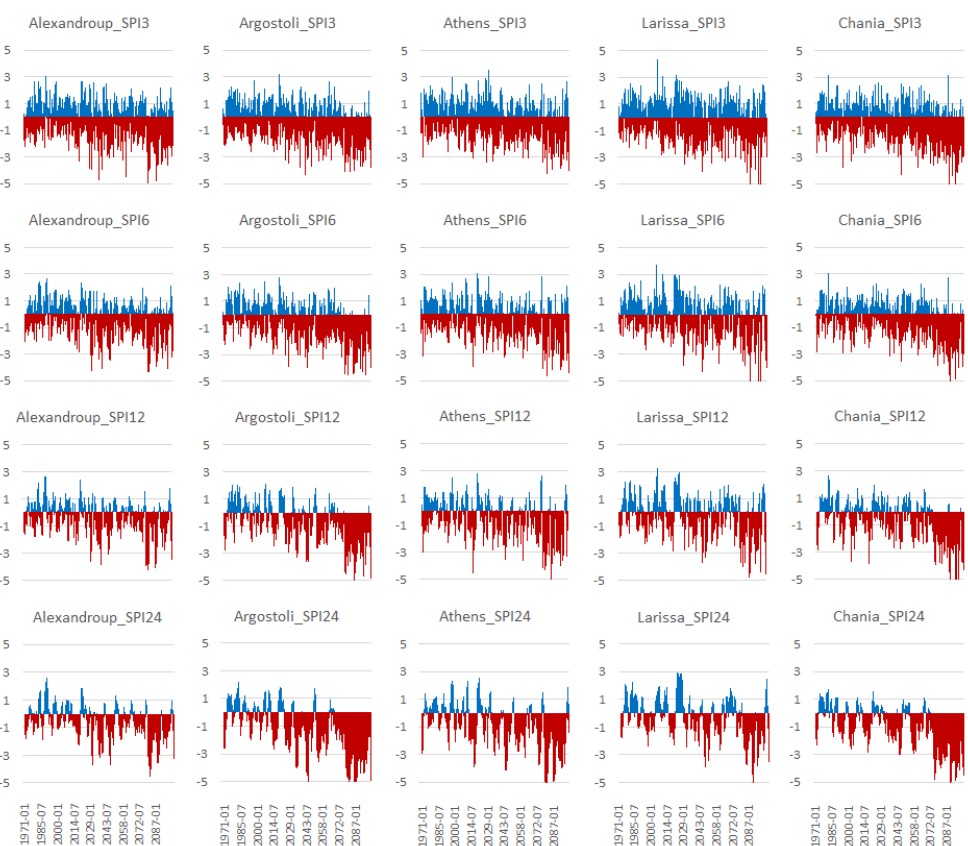

**Figure 14.** Standardized precipitation index estimates for five Greek stations at scales of three (SPI3), six (SPI6), twelve (SPI12), and twenty-four (SPI24) months. The graphs display the temporal variation of SPI over 1971–2100 under RCP8.5.

3.3.2. Temporal Analysis of SPEI-3, SPEI-6, SPEI-12, and SPEI-24

Furthermore, we have calculated the standardized precipitation evapotranspiration index (SPEI). Apart from the temporal precipitation variability, which controls droughts, this index accounts for the effect of temperature variability on droughts. High temperatures enhance evapotranspiration and accelerate drought conditions. As such, during the 2003 European heatwave, the extreme high temperatures dramatically increased evapotranspiration and exacerbated summer drought stress [62]. Therefore, SPEI is preferable for the analysis and assessment of drought risk, especially in areas that are threatened by intense heat and are vulnerable to drought in the future because of climate change.

Similar to Figure 14, the SPEI values for the five representative stations during 1971–2100, under RCP8.5, are presented in Figure 15. The color of each bar indicates positive (green) or negative (orange) SPEI values. The length of the bar indicates the severity of the event. The range of the *y*-axis (−5 to 5) is the same in all subplots to easily compare results. In many cases, we notice the SPEI results exceeding the lower limit of −5. Overall, the combined rising temperatures and precipitation absence lead to large negative SPEI values in all time scales. Therefore, all Greek stations are expected to see changes in the severity and occurrence of droughts in the decades to come during the second half of the 21st century. Based on the SPEI analysis, hydrological droughts of 12 and 24 months are found to persist longer in the future and, in some cases, with greater magnitudes (e.g., Alexandroupoli and Larissa). It is worth noting that the analysis of the two drought indices (SPI and SPEI) reveal significant downward trends in all time scales over the study domain. Additionally, the SPEI index identified more intense drought conditions than the SPI index.

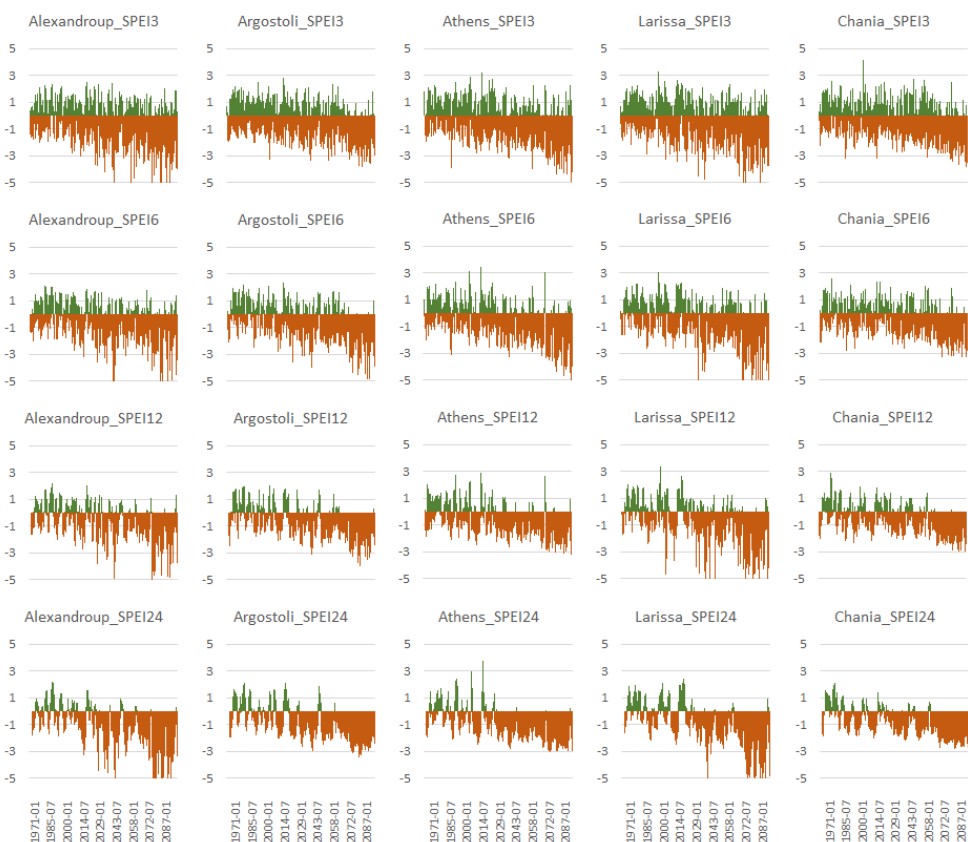

**Figure 15.** Similar to Figure 14, but for the standardized precipitation evapotranspiration index (SPEI).

## 4. Summary and Conclusions

High-resolution simulations from the EURO-CORDEX initiative, under two different representative concentration pathway scenarios, were utilized to calculate several climate indices related to extreme precipitation conditions during the period 1971–2100. First, the model simulations were evaluated against observations over the reference historical period 1971–2000. We utilized several statistical metrics to test the model performance and determine model weaknesses. In general, the climate model was found adequate to estimate the long-term distribution of the variables in all sites.

Consequently, the analysis focused on 21 sites located in areas with varying climatic characteristics across Greece. The Climpact2 software package was run to calculate indicators that are related to hydrological extremes. The analysis revealed changes in the variation of the 14 extreme precipitation indices and, in several cases, statistically significant long-term trends. In the context of future climate change, the results of the trend analysis showed a tendency toward drier conditions. Widespread significant trends were found under RCP8.5. Contrasting trends were found for the dry and wet spells. The consecutive dry days, an index related to meteorological drought, is expected to increase, especially under RCP8.5, with the number of CDD increasing by 20–50% and 40–80% (during 2041–2070 and 2071–2100, respectively). The annual CDD presented upward trends in all cases, but the greater changes are expected in the southern parts of Greece. In contrast, the CWD was seen to decline. The tendency toward a drier climatic future was also supported by the negative trends across the study area in R10mm and R20mm, particularly in the western wet zone, as well as southern Greece. Few significant changes were observed in the Rx1day, Rx5day, R95p, and R99p indices. The PrcpTOT is found to decrease in most parts of the country, with most changes revealed under RCP8.5 and affecting the west, as well as southern Greece. Due to the wet climate in the western regions, the large decrease in precipitation in the western wet zone reflects the impact of climate change in declining rainfall in a place like Greece. In particular, projected changes in total annual precipitation were calculated

as percentage change of the near future (2041–2070) minus present climate (1971–2000) and far future (2071–2100) minus present (1971–2000) for the two RCP scenarios. The medium-term (2041–2070) change is estimated as a 5–20% decrease in precipitation, which may reach a decrease of 10–25% in the long-term (2071–2100) under RCP4.5. Under the high-emission scenario RCP8.5, the annual precipitation is estimated to decrease by 5–25% toward the period 2041–2070 and by 15–40% toward 2071–2100. The R95pTOT indicates small positive trends, indicating areas with higher risk of floods because of increases in the ratio of the sum of all daily precipitation amounts. This study revealed that potential changes in the occurrence and size of droughts can affect the future climate in Greece. Based on the examined scenarios, severe and extreme drought conditions are expected to occur more often based on the future projections. The projection of decreasing temporal trends of SPI indicated a general drying from mid-century and toward the end of the 21st century. SPI-3 was found to be different among seasons. The drought reflected by SPI-3 in spring denoted an earlier arrival of the dry Mediterranean summer. The results obtained from SPI-12 and SPI-24 displayed drying intensification that may cause hydrological and ground water droughts. The results of the SPEI application were generally consistent with SPI, although SPEI revealed a larger increase in dryness compared to SPI. SPEI verified that higher temperatures, which increase evaporation, in combination with a lack of precipitation intensify the risk of droughts. Being sensitive to global warming, SPEI-12 and SPEI-24 showed that the area of study will possibly experience severe droughts in the decades to come. In the Greek region, sectors vulnerable to drought, such as agriculture, tourism, and energy, as well as wildfires, are expected to be heavily affected in the future. A good understanding of the potential threats to water availability in the domain of study is crucial. Hence, the results of this study can be used for monitoring water availability and long-term changes and facilitate planning steps against water scarcity and serious accompanying societal impacts.

**Author Contributions:** Conceptualization, E.K. and C.G.; methodology, validation, formal analysis, visualization, investigation, writing—original draft, E.K.; resources, E.K. and C.G.; writing—review and editing, C.G. All authors have read and agreed to the published version of the manuscript.

**Funding:** This research received no external funding.

**Institutional Review Board Statement:** Not applicable.

**Informed Consent Statement:** Not applicable.

**Data Availability Statement:** Data are available upon reasonable request from authors.

**Acknowledgments:** We acknowledge the European CORDEX (EURO-CORDEX) initiative for producing and making available regional model outputs. We also thank everyone that has contributed to the development of the ClimPACT2 software written in R language and made freely available for the calculation of climate indices.

**Conflicts of Interest:** The authors declare no conflict of interest.

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
