# Peer review of "Projected Changes in Extreme Wet and Dry Conditions in Greece"

_climate, doi:10.3390/cli11030049_

Round 1
Reviewer 1 Report
Overall the manuscript is lengthy by introducing too many indexes. Some indexes can be omitted if other indexes explain them.
Table 1: The following indexes need clear time definition (like per year or annual); CDD, CWD, PRCTOT, R1day, and R5day.
Line 191: SPEI also needs a classification like SPI in Table 2.
Figure 2: Improve the visibility of the map. It is hard to read the stations.
Figure 3: Unit is required for each y axis.
Figure 4 and related text: The model performance is better explained by RMSE than MSE because RMSE has the same unit with the variable used (as author explained). Therefore, MSE can be omitted.
Figure 5: Improve the visibility of graphs. They are hard to read station names, axis values, and axis labels.
Because Table 3 and Table 4 include most results presented in previous figures (from 6 to 12), some figures can be removed without deteriorating the manuscript.
Lines 559-560: Because future climate projections resulted ‘the changes in occurrence and size of droughts’, those change cannot affect future climate conditions. This sentence would be rewritten to clarify causes and effects.
Reviewer 2 Report
The manuscript is interesting, and the results are promising. The authors analyzed the variability of extreme precipitation and drought with different climate indicators. The manuscript needs to be improved prior to its publication. Some points should be improved.
1. Title: It could be more attractive if you chose a balanced word such as “Projected Changes of Drought and Flood in Greece”. Precipitation and drought are not at the same “ level”.
2. Abstract: A summary of the quantitative results should be added to the abstract to highly the finding of your research. The readers could be interested in key quantity results in the abstract parts such as how much percent precipitation and drought change.
3. Introduction: It is more convincing if you clean show the gap or limitation of previous studies compare with yours. What are new findings or what is your contribution compared with previous ones?
4. Methodology: It should give a short introduction to what is EURO-CORDEX and why you choose it instead of others. Could you give more detail on how you "match" the spatial scale and temporal scale of data from RCMs to observational at 21 stations? I am interested in how you computed SPEI in this study. Could you explain more how about it?
5. Results: I guess that the figures were created by Microsoft Excel. It is fine. But you should make it more professional. E.g. the Axis X label should not cover the figure. The Legend should be reorganized. There is a lot of blank space in your figure. Could you explain why Figure 3, Figure 4, and Figure 5 are located out of the result part? The figure should improve the quality. In Figure 13, I think you can use some interpolation techniques such as Invert distance weight or Kriging to present spatial distribution. In Figures 14 and 15, it is hard to view the detail of labels on these figures
6. Summary and Conclusion: In conclusion, it is the better conclusion of extreme drought happens instead of the conclusion of floods. Because your objective is analysis extreme precipitation and drought.

Reviewer 3 Report
This article projected changes in precipitation extremes and particularly to assess drought variability and change across Greece. The author calculated 14 extreme precipitation indices to present precipitation variation, which will help understand potential changes in drought variability under climate change. Before publication in Climate there are some points to revise or explain as listed below.
Provide main results in abstract.
Section 2.4, too many details. Formulas may be concise, compared with long paragraphs of plain text. Also, I think figures are not necessary in method section.
Dose Figure 13 repeat Tables 3 and 4? It would be better to remove Tables to supplementary materials. In Figure 3, dots are less than 21, why?
L147, provide the full name of ETCCDI
